# Five-year follow-up of fundus autofluorescence and retinal sensitivity in the fellow eye in exudative age-related macular degeneration in Japan

Ari Shinojima[1,2], Miki Sawa[3], Ryusaburo Mori[1], Tetsuju Sekiryu[4], Yuji Oshima[5], Aki Kato[6], Chikako Hara[3], Masaaki Saito[4,7], Yukinori Sugano[4], Masayuki Ashikari[6], Yoshio Hirano[6], Hitomi Asato[5], Mayumi Nakamura[8], Kiyoshi Matsuno[8], Noriyuki Kuno[8,9], Erika Kimura[8], Takeshi Nishiyama[10], Mitsuko Yuzawa[1], Tatsuro Ishibashi[5], Yuichiro Ogura[6], Tomohiro Iida[4,11], Fumi Gomi[3,12], Tsutomu Yasukawa[6]*

1 Division of Ophthalmology, Department of Visual Sciences, Nihon University School of Medicine, Tokyo, Japan, 2 Department of Ophthalmology, Keio University School of Medicine, Tokyo, Japan, 3 Department of Ophthalmology, Osaka University Graduate School of Medicine, Suita, Japan, 4 Department of Ophthalmology, Fukushima Medical University School of Medicine, Fukushima, Japan, 5 Department of Ophthalmology, Graduate School of Medical Sciences, Kyushu University, Fukuoka, Japan, 6 Department of Ophthalmology and Visual Science, Nagoya City University Graduate School of Medical Sciences, Nagoya, Japan, 7 Department of Ophthalmology, Graduate School of Medicine and Faculty of Medicine, Akita University, Akita, Japan, 8 Santen Pharmaceutical Co., Ltd., Ikoma, Japan, 9 Japan Innovative Therapeutics, Inc., Nagoya, Japan, 10 Department of Public Health, Aichi Medical University School of Medicine, Nagakute, Japan, 11 Department of Ophthalmology, Tokyo Women's Medical University, Tokyo, Japan, 12 Department of Ophthalmology, Hyogo College of Medicine, Nishinomiya, Japan

* yasukawa@med.nagoya-cu.ac.jp

**Data Availability Statement:** The data used to support the findings of this study are restricted by the institutional ethics committees at Nagoya City

## Abstract

### Purpose

To assess the 5-year change in abnormal fundus autofluorescence (FAF) patterns and retinal sensitivity in the fellow eye of Japanese patients with unilateral exudative age-related macular degeneration (AMD).

### Methods

Patients with unilateral exudative AMD who developed abnormal FAF in the fellow eyes were enrolled. FAF imaging and microperimetry were performed at baseline and follow-ups. FAF findings were classified into 8 patterns based on the International Fundus Autofluorescence Classification Group to assess retinal sensitivity. Forty-five points covering the central 12 degrees on microperimetry were superimposed onto the FAF images. Each point was classified depending on the distance from the abnormal FAF. "Close" was defined as the portion within 1 degree from the border of any abnormal FAF, and "Distant" was defined as the portion over 1 degree from the border of abnormal FAF. To investigate the association between the retinal sensitivity and distance from the abnormal FAF, hierarchical linear mixed-effect models were used with the distance, time and time squared from baseline

University Graduate School of Medical Sciences, Nihon University School of Medicine, Osaka University Graduate School of Medicine, Fukushima Medical University School of Medicine, and Kyushu University Graduate School of Medical Sciences in order to protect patient privacy. Data are available from Clinical Research Management Center, Nagoya City University Hospital (e-mail: ctmc@med.nagoya-cu.ac.jp) for researchers who meet the criteria for access to confidential data.

**Funding:** Santen Pharmaceutical Co., Ltd. provided support in the form of honoraria for speaking and/ or organizing at meetings (AS, MS, RM, TS, Yuji Oshima, TI, Yuichiro Ogura, FG, TY), but did not have any additional role in the study design, data analysis, decision to publish, or preparation of the manuscript. Santen Pharmaceutical Co., Ltd. provided support in the form of salaries for authors [MN, KM, NK, and EK], but did not have any additional role in the study design, data analysis, decision to publish, or preparation of the manuscript. The specific roles of these authors are articulated in the 'author contributions' section. This work was also supported by JSPS KAKENHI Grant Number JP19K18893.

**Competing interests:** Santen Pharmaceutical Co., Ltd. provided support in the form of honoraria for speaking and/or organizing at meetings [TY, RM, MS, AS, TS, Yuji Oshima, TI, Yuichiro Ogura, TI, FG]. Mayumi Nakamura, Kiyoshi Matsuno, Noriyuki Kuno, and Erika Kimura are (were) employees of Santen Pharmaceutical Co., Ltd. This does not alter our adherence to PLOS ONE policies on sharing data and materials.

(months), and angle (degrees) as fixed effects. Differences among patients, eyes, and test point locations were considered successively nested random effects.

## Results

We studied 66 fellow eyes with abnormal FAF. Twenty-seven eyes were followed-up during the 5 years. In the 13 of 27 eyes (48%), the abnormal FAF patterns had changed during the 5 years. We found retinal sensitivity was associated significantly with the distance from the abnormal FAF ("Distant": p<0.001, time$^2$ from baseline: p<0.001, angle: p<0.001). The mean retinal sensitivity of the "Close" tended to deteriorate after the third year and eventually showed the similar sensitivity as the portion within the abnormal FAF.

## Conclusion

FAF patterns can change about half during the 5 years and the retinal sensitivity near abnormal FAF tends to deteriorate after the third year.

## Introduction

Age-related macular degeneration (AMD) is a progressive retinal degenerative disease in the elderly [1]. There are several classifications of AMD in the literature [2]. In this study, we use the classification according to the Age-Related Eye Disease Study (AREDS) category [3]. According to the AREDS, no AMD (AREDS category 1) represents the control group; it is characterized by no or few small drusen (<63 μm in diameter). Early AMD (AREDS category 2) is characterized by a combination of multiple small drusen, few intermediate drusen (63–124 μm in diameter), or mild RPE abnormalities. Intermediate AMD (AREDS category 3) is characterized by any of the following features: numerous intermediate drusen, at least one large drusen (>125 μm in diameter), or geographic atrophy (GA). Advanced AMD (AREDS category 4) is characterized by one or more of the following in one eye: GA of the RPE involving the foveal center and neovascular maculopathy. Previous studies reported that the fellow eye in unilateral exudative AMD represents a risk factor for the development of choroidal neovascularization (CNV) or GA involving the center of the macula [4–6]. In intermediate-to-late AMD, patients are recommended to quit smoking [7,8] and maintain a balanced and healthy diet [9]. Other risk factors such as hypertension, atherosclerosis, overweight [10], and genetic factors [11] are also reported. Therefore, the detection of early-to-intermediate AMD is important for the prophylactic guidance to patients. Microperimetry is a useful tool to detect a slight functional change and fundus autofluorescence (FAF) is a useful tool to detect a slight fundus change non-invasively.

Many studies related to microperimetry have been performed [12–14]. Clinically, microperimetry is used for diseases with visual impairment, such as AMD [15,16], central serous chorioretinopathy [17,18], angioid streaks [19], and retinal dystrophy including choroideremia [20,21].

FAF imaging with a short wavelength detects the lipofuscin distribution in the retinal pigment epithelium (RPE) cell monolayer noninvasively [22,23]. Therefore, FAF imaging supports a quantifiable assessment of GA through detection of a hypofluorescent atrophic area [24, 25]. In addition, various abnormal FAF patterns have been reported in the eyes with early-

to-intermediate AMD. Our previous study demonstrated that the prevalence of abnormal FAF patterns and clinical features of late AMD were different among races [9].

The association between microperimetry and FAF imaging has been reported with a short-duration follow-up [9,26]. We previously reported the one-year results [9]. However, long-duration follow-ups and the clinical course of retinal sensitivity and FAF pattern changes have not been reported.

Our aim is to demonstrate the clinical course of retinal sensitivity and FAF pattern changes without treatment in the fellow eye of patients with unilateral exudative AMD in a Japanese population with a long-duration follow-up.

## Materials and methods

### Study design

The Japanese Fundus Autofluorescence and Microperimetry in Early Age-Related Maculopathy (JFAM) study group conducted this study from December 2006 to March 2014 at 5 university hospitals in Japan. Data of all consecutive patients with unilateral exudative AMD who developed abnormal FAF in the fellow eyes were prospectively studied.

### Ethical statement

This study adhered to the tenets of the Declaration of Helsinki. The Institutional Ethics Committees of the Nagoya City University Graduate School of Medical Sciences, Nihon University School of Medicine, Osaka University Graduate School of Medicine, Fukushima Medical University School of Medicine, and Kyushu University Graduate School of Medical Sciences reviewed and approved the study protocol (University Hospital Medical Information Network approval number: R000043372/UMIN000038050). Patients provided written informed consent before participating in the study.

### Study patients and protocol

The eligible patients were Japanese, aged ≥50 years, and had unilateral advanced AMD with exudative choroidal neovascularization. The inclusion criterion was hyper- or hypofluorescence on FAF imaging in the fellow eye without exudative choroidal neovascularization. FAF imaging was performed with the Heidelberg Retina Angiogram Digital Angiography System (HRA) or HRA2 (Heidelberg Engineering, Heidelberg, Germany). The major exclusion criteria were the presence of exudative findings in the fellow eyes, including CNV, hemorrhage, serous RPE detachment, serous retinal detachment, and hard exudates, diabetic retinopathy, uveitis, high myopia (less than -8.0 diopters), retinal vein occlusion, hazy media interfering with fundus examinations, or a history of laser photocoagulation. In this study, the fellow eyes had drusen and/or pigmentary abnormalities (n = 65) and extrafoveal geographic atrophy (n = 1) at initial visit [9].

The participants were monitored during the 5 years. Each participant underwent measurements of the best-corrected visual acuity (BCVA), binocular funduscopy, and color fundus photography at baseline and at the 3, 6, 9, 12, 24, 36, 48, and 60 months. FAF imaging and microperimetry were performed at baseline and at the 6, 12, 24, 36, 48 and 60 months.

Macular sensitivity was measured with a microperimetry device (MP-1, Nidek Technologies, Padova, Italy) using the Nidek Advanced Vision Information System software (NAVIS; Nidek Technologies). We adopted the same parameters as previously reported, a customized radial grid of 45 stimuli covering the central 12 degrees, stimuli size Goldman III with a

presentation time of 200 ms, and 0–20 dB stimulus light intensity of 4-asb white light as the background. A 4–2 stimuli strategy was used [9].

## Image analysis

FAF patterns were determined based on the classification system provided by the International Fundus Autofluorescence Classification Group (IFAG) [27,28]. Four authors (ST, MS, YO, and TY) graded the images into the following 8 patterns: minimal change, focal increase, focal plaque-like, patchy, linear, lace-like, reticular, and speckled. The JFAM study group created a flow chart to facilitate the pattern classification, which was based on the IFAG definition [9]. We used the flow chart to evaluate the FAF patterns.

The decision of the area on FAF was made by one retina expert (TY) for all cases. Hyperfluorescent and/or hypofluorescent portions were demarcated manually. Subsequently, these images were overlaid on fundus photographs to assess colocalization of the hyperfluorescent and/or hypofluorescent spots with the findings on color funduscopy. The baseline retinal sensitivity maps on microperimetry were overlaid on the baseline FAF images using the dedicated Nidek Advanced Vision Information System software. Retinal points tested with microperimetry were classified into 3 groups at baseline based on the distance from the abnormal FAF: "Within" (within abnormal FAF), "Close" (within 1 degree from the border of abnormal FAF), and "Distant" (over 1 degree from abnormal FAF). Those test patches were followed-up until the 5th year with baseline classifications. The mean retinal sensitivity in each group was calculated at each time point. The average change from baseline to each time point was also assessed.

## Statistical analysis

The paired t-test was used to compare the retinal sensitivity at each year compared to baseline by each pattern of abnormal FAF during the 5 years. $P < 0.05$ was considered significant.

To investigate the association between the retinal sensitivity and distance from the abnormal FAF, hierarchical linear mixed-effect models were used with the distance, time and time squared from baseline (months), and angle (degrees) as fixed effects. Differences among patients, eyes, and test point locations were considered successively nested random effects, i.e., test point locations nested within eyes within patients.

We used two models, the first of which included a random intercept for each patient, each eye nested within each patient, and each test point location nested within each eye within each patient (model 1). The second model added random slopes to time and time squared from baseline (model 2). Akaike's information criterion (AIC) and Bayesian information criterion (BIC) were used to compare the statistical models used. Both criteria are parsimony-adjusted indices useful for examining the fit of competing models. These criteria are based on the log likelihood value of a given model and impose a penalty on over-parameterized models. With these statistics, the preferred model is associated with lower relative values [28].

Analysis was performed with the nlme package in R [Jose Pinheiro DB, DebRoy Saikat, Sarkar Deepayan, the R Core team (2018). nlme: Linear and Nonlinear Mixed Effects Models].

## Results

### Patient characteristics

Sixty-six fellow eyes with abnormal FAF were enrolled in this study. The baseline color fundus photographs of 62 eyes and FAF images of 66 eyes were of sufficient quality [9]. No eyes with a normal pattern were enrolled in this study. Exudative AMD occurred in 14 (21.2%) of 66 eyes

during the 5 years in fellow eyes, with the occurrence in 6 eyes within 1 year and in 8 eyes between the 2- and 5-year follow-ups. Advanced AMD such as polypoidal choroidal vasculopathy (PCV), typical AMD, and retinal angiomatous proliferation were detected in 4, 9, and 1 eye, respectively (the six eyes are written in Table 4 of our previous report [9], and the other 8 eyes are written in Table 1 of this manuscript).

Of the remaining 52 eyes, the number of patients' loss to follow-up was 2, 5, 6, 3, and 3 at the 1-, 2-, 3-, 4-, and 5-year follow-ups. Thirty-three eyes completed the 5-year follow-up. However, six eyes were followed-up with only microperimetry or FAF. Twenty-seven eyes completed 5 years of follow-up with both microperimetry and FAF imaging (right eyes were 12 (44.4%) (Table 2).

## Fundus findings and patterns of abnormal FAF

FAF patterns of sixty-six eyes when enrolling had 8 patterns. However, 27 eyes which were followed-up during the 5 years had only 7 patterns. Twenty-seven eyes visualized on microperimetry at baseline were classified into the following 7 patterns: minimal change 1, focal increase 10, linear 2, focal plaque-like 4, patchy 6, lace-like 3, and speckled 1 (Fig 1).

## Transition to other FAF patterns

In 13 of 27 eyes (48%), the abnormal FAF patterns changed at the 5th year (Fig 1 and Table 3).

## Visual acuity and retinal sensitivity

The mean baseline logarithm of the minimum angle of resolution (logMAR) BCVA of 66 eyes was -0.011 ± 0.128. The mean logMAR BCVA of the 27 eyes that underwent both microperimetry and FAF decreased from -0.023 ± 0.108 at baseline to 0.058 ± 0.188 at the 5th year, which showed significance (p = 0.026, paired t-test).

The retinal sensitivity within 12 degrees was measurable during the 5 years in 27 eyes (1,215 points on microperimetry in total, others were not eligible for analysis). The baseline mean retinal sensitivity was 14.1 ± 4.3 dB, which decreased to 13.5 ± 5.8 dB at the 5th year. The mean baseline retinal sensitivity of the 14 eyes that developed AMD was 12.9 ± 4.4 dB (630 points in total). The mean retinal sensitivity in eyes with each abnormal FAF pattern differed among the abnormal FAF patterns. The speckled FAF pattern had the lowest retinal sensitivity at the 5th year. Although linear, focal plaque-like, patchy and speckled FAF pattern deteriorated at the 5th year compared to baseline, lace-like pattern improved at the 5th year (Fig 2).

**Table 1. Characteristics of eight eyes with progression to exudative AMD.**

| Case | Age | Month with withdrawal | Mean retinal sensitivity | | Abnormal FAF pattern | Hard drusen | Soft drusen | Confluent drusen | Hyper pigmentation | Hypo pigmentation |
|------|-----|----------------------|---------|------------------|---------------------|-------------|-------------|------------------|--------------------|-------------------|
| | | | Baseline | Final Examination | | | | | | |
| 1 | 88 | 17 | 14.0 | 14.0 | Patchy | + | + | - | + | - |
| 2 | 72 | 35 | 15.6 | 8.5 | Patchy | + | + | + | - | - |
| 3 | 81 | 30 | 12.3 | 17.5 | Patchy | - | + | + | - | - |
| 4 | 69 | 42 | 16.1 | 11.8 | Patchy | - | + | + | - | - |
| 5 | 80 | 14 | 13.9 | 16.6 | Minimal change | + | + | - | - | + |
| 6 | 80 | 22 | 10.1 | 12.4 | Reticular | - | + | + | + | - |
| 7 | 80 | 39 | 9.7 | 7.4 | Focal increase | - | + | + | + | - |
| 8 | 77 | 26 | 12.8 | 14.4 | Patchy | - | + | + | - | - |

**Table 2. Baseline profiles of study patients (n = 27).**

| | |
|---|---|
| Age (mean ± SD, years) | 71.2 ± 7.2 (54–85) |
| Gender | |
| Male | 19 (70.4%) |
| Female | 8 (29.6%) |
| Best-Corrected Visual Acuity, logMAR (Snellen) | -0.02 ± 0.11 (20/20) |
| Mean retinal sensitivity | 14.1 ± 4.3 |
| Type of drusen—number of eye (%) (Some findings are overlapped) | |
| Hard drusen | 7 (25.9%) |
| Soft drusen [with confluent] | 25 (92.6%) [12 (44.4%)] |
| Pigmentary abnormalities | 16 (59.3%) |
| Diagnosis of fellow eye—no. (%) | |
| Typical neovascular AMD | 19 (70.4%) |
| PCV | 8 (29.6%) |

SD: standard deviation; LogMAR: logarithm of the minimum angle of resolution.

## Retinal sensitivity and the distance

In total, 1,215 points on microperimetry in 27 eyes were followed-up during the 5 years and were classified into 3 groups (Within, Close, and Distant) based on the distance from the abnormal FAF. The mean retinal sensitivities were 15.0 ± 0.8, 12.3 ± 0.9 and 10.6 ± 0.9 dB in the Distant, Close and Within group, respectively, at baseline. "Close" showed deterioration in sensitivity after the 3rd year (p < 0.05) compared to baseline and resulted in 10.7 ± 1.3 dB at the 5-year follow-up. The mean retinal sensitivity of the "Close" tended to deteriorate after the

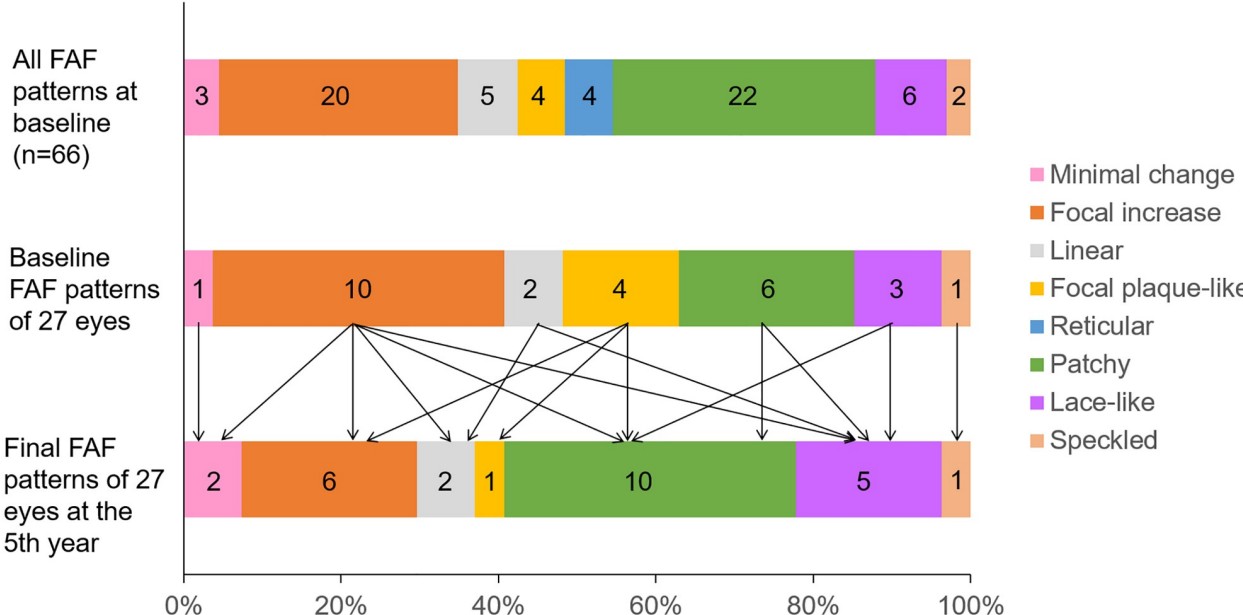

**Fig 1. The proportions of FAF patterns.** Top: all FAF patterns of 66 eyes at baseline. Middle: FAF patterns at baseline of 27 eyes which were followed-up during the 5 years. Bottom: the final FAF patterns of 27 eyes at the 5th year. The numbers within graphs show the actual number of the eye.

**Table 3. Pattern alterations observed during follow-up.**

| Initial FAF pattern | Last FAF pattern | Number of eyes, n (%) |
|---|---|---|
| Focal increase | minimal change | 1 (7.7%) |
| | linear | 1 (7.7%) |
| | patchy | 2 (15.4%) |
| | lace-like | 2 (15.4%) |
| Linear | lace-like | 1 (7.7%) |
| Focal plaque-like | focal increase | 2 (15.4%) |
| | patchy | 1 (7.7%) |
| Patchy | lace-like | 1 (7.7%) |
| Lacelike | patchy | 2 (15.4%) |
| Total | | 13 (100%) |

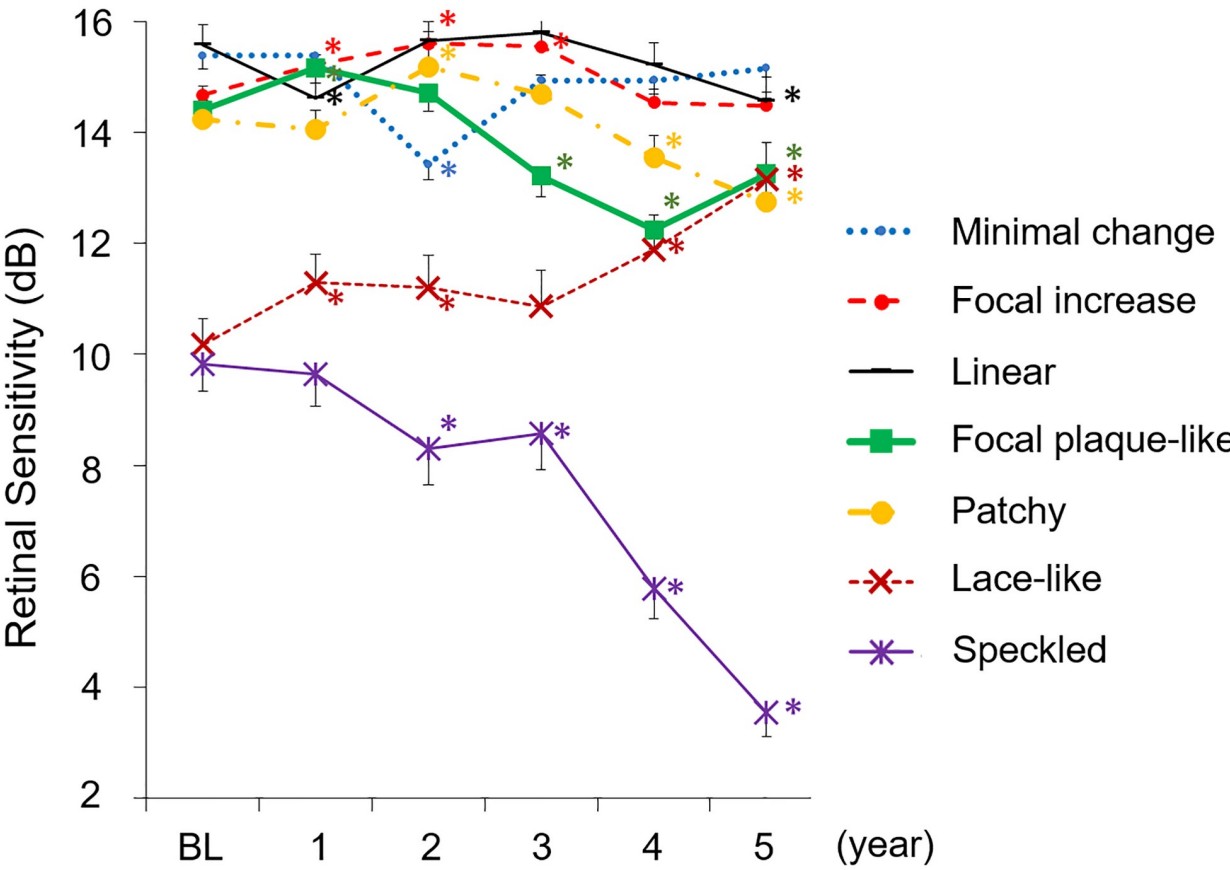

**Fig 2. Transition of mean retinal sensitivity by each pattern of abnormal FAF during the 5 years.** The number of each pattern differs at each time-point compared to baseline (cf. Fig 1). The sensitivity of the linear, focal plaque-like, patchy, and speckled FAF pattern decreased significantly compared to the baseline at the 5th year. The speckled FAF pattern at the 5th year had the lowest mean retinal sensitivity. The sensitivity of lace-like increased significantly compared to baseline at the 5th year. Data are expressed as mean ± standard error (SE). * P < 0.05 compared to baseline, paired t-test. BL: baseline.

third year and showed eventually the similar sensitivity as the portion within the abnormal FAF (Fig 3).

First, we fitted a random intercept model to account for a three-level nested data structure. i.e., test point locations nested within eyes within patients (model 1). We also fitted a random intercept and slope model to assess possible variation on the effect of time and time squared from baseline across patients, eyes nested within patients and test point locations nested within eyes within patients (model 2). In a comparison of the two models, model 2, provided a better fit than model 1, with an AIC and BIC of 30897.95 and 30964.84 in model 1, and 30155.77 and 30262.79 in model 2, respectively. Thus, we examined the association between the retinal sensitivity and the distance from the abnormal FAF based on model 2 and found the retinal sensitivity was associated significantly with the distance from the abnormal FAF (Table 4).

Fig 4 is a representative case of FAF imaging and color fundus imaging.

## Discussion

In the current study, we studied 66 eyes with unilateral exudative AMD, which showed an abnormal FAF in the contralateral eye. In total, 14 out of 66 eyes (21.2%) progressed to exudative AMD during the 5-year follow-up. Patients with advanced AMD or vision loss due to non-advanced AMD in 1 eye (Category 4) reportedly have a 43% expected probability of progression to advanced AMD in the fellow eye after 5 years [6]. CNV reportedly develops in the other eye in 12.3% of the Japanese population by 5 years [5]. Our result showed a higher incidence than that reported 20 years before in Japan. This may be influenced by westernization of the diet.

In this study, seven (50%) of 14 eyes which progressed to exudative AMD during the 5 years showed the patchy FAF pattern. The remaining patchy FAF pattern (6 eyes (22.2%) at

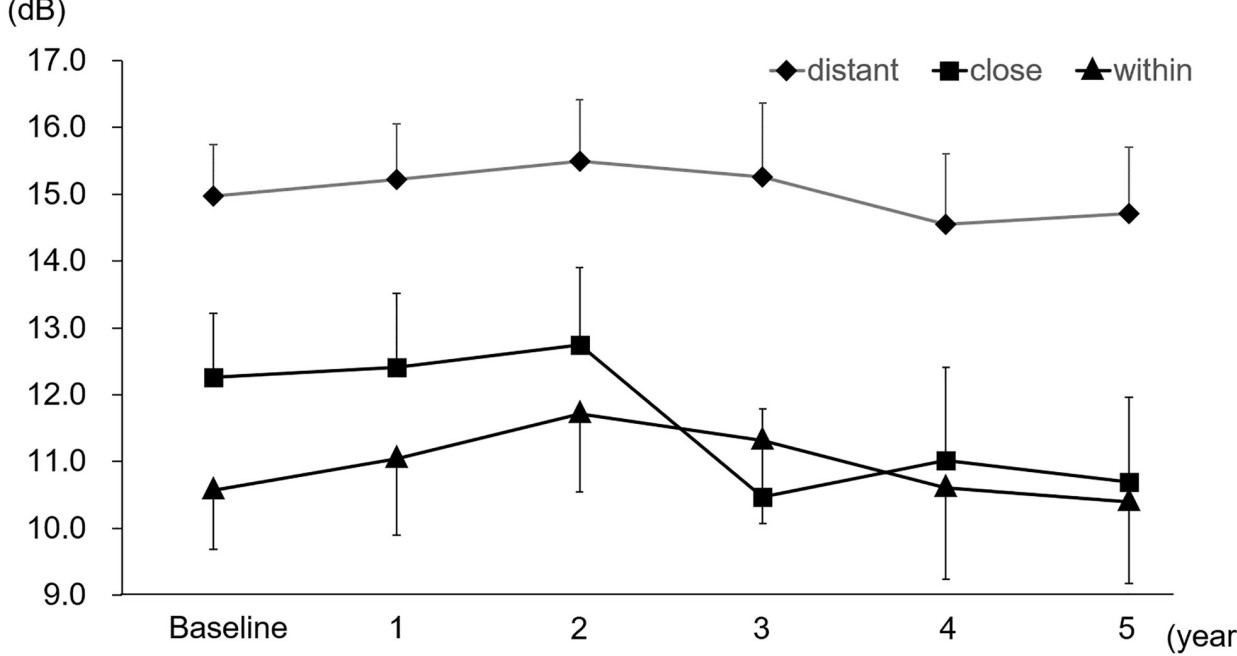

**Fig 3. The association between the mean retinal sensitivity and the distance from the abnormal FAF (1,215 points per year).** Total 1,215 points were classified as "Within" (within abnormal FAF), "Close" (within 1 degree from the border of abnormal FAF), and "Distant" (over 1 degree from the border of abnormal FAF). Data are expressed as mean ± SE.

**Table 4. Results of the random intercept and slope model.**

|  | Estimate | SE | *P*-value |
|---|---|---|---|
| Distance: Within | Reference | | |
| Close | 0.48 | 0.40 | 0.227 |
| Distant | 2.34 | 0.37 | <0.001 |
| Time | 0.04 | 0.01 | <0.001 |
| Time$^2$ | <-0.01 | <0.01 | <0.001 |
| Angle | 0.41 | 0.04 | <0.001 |

baseline) showed no significant change in retinal sensitivity until the 3rd year but showed a significant decrease in retinal sensitivity at the 3rd, 4th and 5th year compared to baseline (10 eyes (37.0%) at the 5th year) (Fig 2). Although the retinal sensitivity of patchy, linear, focal plaque-like and speckled pattern deteriorated at the 5th year compared to baseline, the lace-like pattern improved at the 5th year in this study. The mechanism of recovery of retinal sensitivity is unknown. It is possible that the lace-like pattern may be associated with RPE hyperplasia or other transient stressed condition. Because the lace-like pattern was not co-localized with drusen. In our previous report, we demonstrated that antioxidant supplementation might affect the recovery of retinal sensitivity [9]. Therefore, supplementation and smoking cessation may

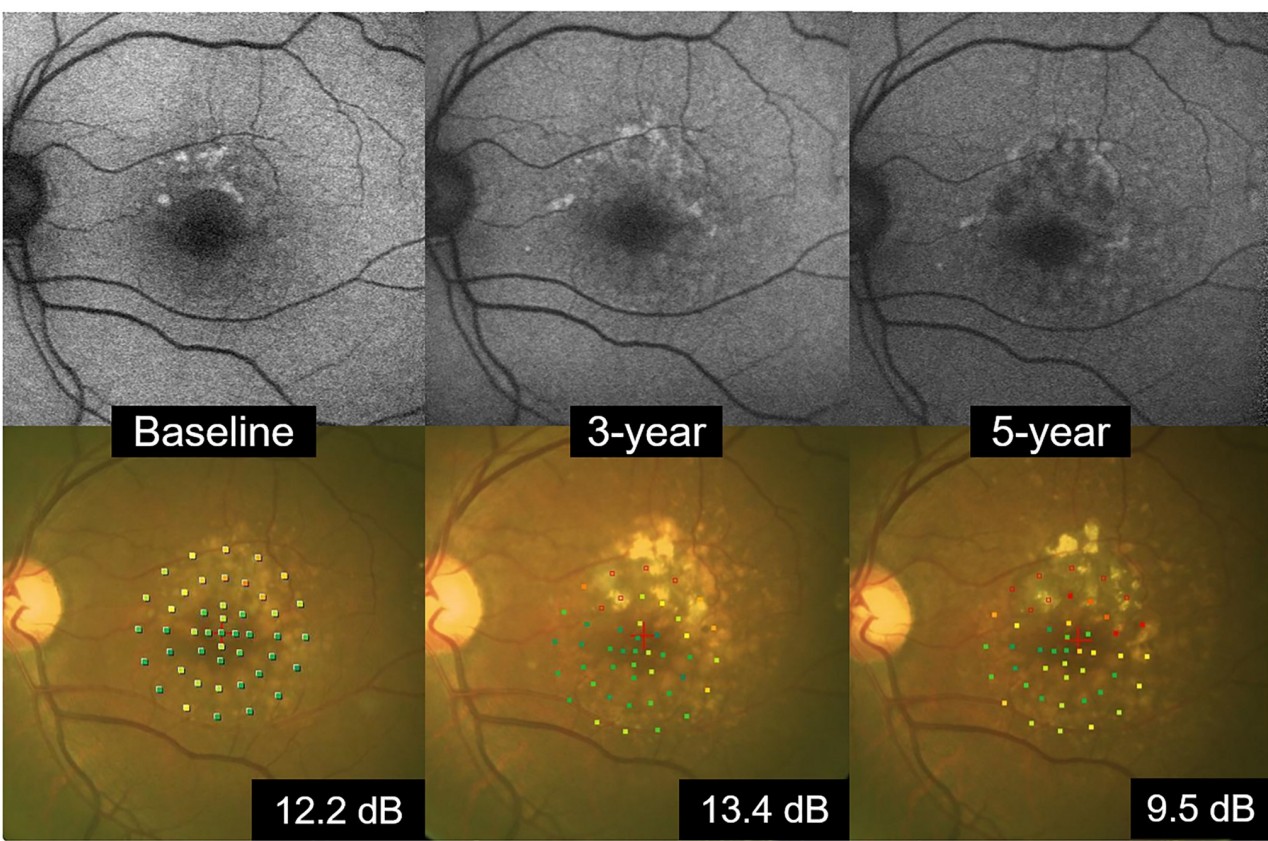

**Fig 4. A representative case of FAF imaging and color fundus imaging.** A 66-year-old man. The focal increase FAF pattern was seen at baseline, and the mean retinal sensitivity was 12.2 dB. In 3 years, the portion showing hyperfluorescence at baseline changed to hypofluorescence in part, and the hyperfluorescence expanded and changed to the patchy pattern. In the fundus photograph, drusen increased during the 5 years. The mean retinal sensitivity was 13.4 dB at the 3$^{rd}$ year and 9.5 dB at the 5th year.

be related to the recovery of this specific pattern of abnormal FAF. Possible other reasons may involve the learning effect and the inaccuracy of auto-tracking of the microperimeter.

FAF pattern changes in this study was partially similar to other report [29]. Further study is needed to ascertain whether these pattern changes are reversible or not.

In general, retinal sensitivity in areas showing an abnormal FAF pattern was low from the start. In this study, test patches within 1 degree from the border of an abnormal FAF pattern deteriorated during the 5-year course. Retinal sensitivity decreased significantly in the Close group after the 3-year follow-up compared to baseline, probably as a result of the expansion of abnormal FAF with time. The test patches which were "Close" group may change to "Within" group during the 5 years. This indicates that a more careful follow-up is necessary for the "Close" (within 1 degree from the border of abnormal FAF) group. On the other hand, areas over 1 degree from the border of an abnormal FAF pattern showed good retinal sensitivity during the 5-year course.

This study has some limitations, including the absence of systematic evaluation of other known risk factors for AMD such as hypertension, atherosclerosis, overweight [10], and genetic factors [11]. Some older subjects may get weaker and less cognitive during the 5-year follow-up period even though they are perfectly healthy at the time of enrollment. An age-matched control group is necessary in the future study.

## Conclusions

The fellow eyes of patients with unilateral exudative AMD were studied to assess the 5-year change. Our results imply that FAF patterns can change about half during the 5-year and the retinal sensitivity within 1 degree from the border of abnormal FAF tends to deteriorate after the third year. Microperimetry and FAF are presumed to be useful for long-term changes in retinal sensitivity and prediction of prognosis.

## Acknowledgments

The authors thank Editage (www.editage.com) for English language editing.

## Author Contributions

**Conceptualization:** Tsutomu Yasukawa.

**Data curation:** Ari Shinojima, Miki Sawa, Ryusaburo Mori, Tetsuju Sekiryu, Yuji Oshima, Aki Kato, Chikako Hara, Masaaki Saito, Yukinori Sugano, Masayuki Ashikari, Yoshio Hirano, Hitomi Asato, Mayumi Nakamura, Kiyoshi Matsuno, Noriyuki Kuno, Erika Kimura, Tomohiro Iida, Fumi Gomi, Tsutomu Yasukawa.

**Formal analysis:** Takeshi Nishiyama, Tsutomu Yasukawa.

**Funding acquisition:** Ari Shinojima.

**Investigation:** Ari Shinojima, Miki Sawa, Ryusaburo Mori, Tetsuju Sekiryu, Tsutomu Yasukawa.

**Methodology:** Tsutomu Yasukawa.

**Supervision:** Tetsuju Sekiryu, Mitsuko Yuzawa, Tatsuro Ishibashi, Yuichiro Ogura, Tomohiro Iida, Fumi Gomi.

**Validation:** Ari Shinojima, Miki Sawa, Ryusaburo Mori, Tetsuju Sekiryu, Yuji Oshima, Aki Kato, Chikako Hara, Masaaki Saito, Yukinori Sugano, Masayuki Ashikari, Yoshio Hirano,

Hitomi Asato, Mitsuko Yuzawa, Tatsuro Ishibashi, Yuichiro Ogura, Tomohiro Iida, Fumi Gomi, Tsutomu Yasukawa.

**Writing – original draft:** Ari Shinojima.

**Writing – review & editing:** Tsutomu Yasukawa.

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
