## [Decision Letter · Decision Letter 0]

30 Oct 2019

PONE-D-19-28095

Five-Year Follow-Up of Fundus Autofluorescence and Retinal Sensitivity in the Fellow Eye in Age-Related Macular Degeneration in Japan

PLOS ONE

Dear Dr. Yasukawa

Thank you for submitting your manuscript to PLOS ONE. After careful consideration, we feel that it has merit but does not fully meet PLOS ONE’s publication criteria as it currently stands. Therefore, we invite you to submit a revised version of the manuscript that addresses the points raised during the review process.

We would appreciate receiving your revised manuscript by December 31, 2109. To enhance the reproducibility of your results, we recommend that if applicable you deposit your laboratory protocols in protocols.io, where a protocol can be assigned its own identifier (DOI) such that it can be cited independently in the future. For instructions see: http://journals.plos.org/plosone/s/submission-guidelines#loc-laboratory-protocols

A rebuttal letter that responds to each point raised by the academic editor and reviewer(s). This letter should be uploaded as separate file and labeled 'Response to Reviewers'. You do not need agree with all the points raised by the reviewers.A marked-up copy of your manuscript that highlights changes made to the original version. This file should be uploaded as separate file and labeled 'Revised Manuscript with Track Changes'.An unmarked version of your revised paper without tracked changes. This file should be uploaded as separate file and labeled 'Manuscript'.

We look forward to receiving your revised manuscript.

Kind regards,

Yuhua Zhang

Academic Editor

PLOS ONE

2. We understand this manuscript may be closely relate to a previous publication, written by you/you co-authors:

Fundus autofluorescence and retinal sensitivity in fellow eyes of age-related macular degeneration in Japan.

https://doi.org/10.1371/journal.pone.0213161

If the participant populations of the two studies overlap please include this information in the Methods section.

"Santen Pharmaceutical Co., Ltd. provided support in the form of honoraria for speaking and/or organizing at meetings (AS, MS, RM, TS, Yuji Oshima, TI, Yuichiro Ogura, FG, TY), but did not have any additional role in the study design, data analysis, decision to publish, or preparation of the manuscript. This work was also supported by JSPS KAKENHI Grant Number JP19K18893.".

We note that you received funding from a commercial source: [Santen Pharmaceutical Co., Ltd]

Reviewers' comments:

Reviewer's Responses to Questions

**Comments to the Author**

1. Is the manuscript technically sound, and do the data support the conclusions?

Reviewer #1: Partly

Reviewer #2: Yes

Reviewer #3: Partly

2. Has the statistical analysis been performed appropriately and rigorously? 

Reviewer #1: N/A

Reviewer #2: No

Reviewer #3: I Don't Know

3. Have the authors made all data underlying the findings in their manuscript fully available?

Reviewer #1: Yes

Reviewer #2: Yes

Reviewer #3: Yes

4. Is the manuscript presented in an intelligible fashion and written in standard English?

Reviewer #1: Yes

Reviewer #2: Yes

Reviewer #3: Yes

5. Review Comments to the Author

Reviewer #1: 1. P9 line 137: What do you mean “hyper- or hypofluorescence on FAF imaging in the fellow eye”? The follow eye has hypofluorescence but not (dry) AMD? Or it may has (dry) AMD also? If both eye has been diagnosed to have AMD, which eye would be considered as fellow eye? Although it has been stated in the previous study in the series, it is still worth a brief description in this paper.

2. Figure 1: standard deviation bar is missing. Also mark how many eyes and percentage of the 27 included for each pattern.

3. Figure 2: mark how many spots and also percentage of total analyzed spots for distant, close and within.

4. Sadly the only FAF pattern (speckled) shows worst progression and no pattern change has only 1 eye under study. Though the study cannot be re-done, yet the conclusion may not be convincing based on this very small sample size.

5. P19 Line293 In terms of the pattern change happened in this study during the 5 year, there is a brief description in the discussion but including a graph could be more helpful. How’s your pattern change trends compared to other studies?

6. P20 Line303 How would you explain the “Close” pattern the sensitivity slightly “recovers” between 3 and 5 years? Worth more discussion.

7. P20 Line316 The reversible sensitivity also appears in lace-like pattern, what do you (and others) say about it?

Reviewer #2: This prospective, observational study assessed the 5-year changes of FAF and retinal sensitivity evaluated by microperimetry in the fellow eyes of Japanese patients with unilateral

nAMD. I have the following questions:

1. Line 205-210, in this paragraph the authors described the data of the 14 eyes which progressed to nAMD (6 within Year 1 and 8 between Year 2 to Year 5), but in Table 2 the authors listed only the data of the 8 eyes progressed between Year 2 to Year 5, which made some data appear to be unmatched and confusing. Please clarify this.

2. Have the authors tried to compare the demographics, the FAF patterns, mean retinal sensitivity, as well as the % of “within”, “close”, and “distant” spots between the 8 eyes progressed to nAMD between 2-5 years and the other 6 eyes progressed within 1 year?

3. The exclusion criteria based on refractive error was < -8D, which was looser than most studies appoint -6D as the diagnostic criteria of high myopia. Previous studies showed macular sensitivity decreases with spherical equivalent. Given that East Asian countries have more high myopic patients than anywhere else over the world, I have a concern that if too many subjects in this study had relatively high myopia (between -6D to -8D), the retinal sensitivity results could be biased. Please comment on this point and if possible, exclude those with relatively high myopia.

4. Line 218-222, is there any statistical significance between the baseline mean sensitivity of the 27 eyes without progression and the 14 eyes with progression, and between the baseline and Year 5 mean sensitivity of the 27 eyes? What’s the sensitivity of the 14 eyes at the endpoint? Or the last measurement before those eyes became wet?

5. Line 238-239, “…deteriorated after 3 years compared to the other patterns (p < 0.001). *P < 0.05 compared to baseline, paired t-test.” Can the authors clarify what’s the statistical method used and how the comparisons were made here? Is it ever possible to use paired t-test to do this comparison (speckled VS other patterns)? Similarly, in Line 246-247, the method for conducting the statistical analysis should also be clearly stated in the Methods Section.

6. With the disease progression during the 5 years, I think it is likely that the prior “distant” point may later turn “close” or “within”, and the prior “close” point may later become “within”. If such alteration happened, how were the points classified? In Figure 2, the authors mentioned that paired t test was used. Does it mean that even a point has altered over time, it still belongs to the initial group?

7. Is it possible to divide each group (within, close, and distant) into 2 subgroups: progressive and non-progressive, and compare their sensitivity?

8. Line 252-255, unfortunately, the statistical methods were inappropriate no matter whether it was paired or unpaired t test, as the authors described in the Methods Section. When analyzing the 1215 points obtained from 27 eyes during multiple follow-ups, the data was not independent (multiple points from one eye) and repeated measurements were taken. Therefore, a mixed-effects model or a generalized estimating equation should be used. Also, the deterioration rate of the retinal sensitivity can be modeled by this means.

9. The test statistics (e.g., F, t) should be shown before the P value.

10. Line 266-269, please clearly describe how many cases have changed from one pattern to another, with more detailed demographic features and the converted time of those eyes. A table is preferred.

11. The authors stated that the lace-like FAF pattern may be associated with retinal sensitivity reservation over time. However, there was only one such case involved in the study, and the slight increase of retinal sensitivity at Year 3 can also be attributed to the learning effect of the patient. More careful investigations should be conducted before this conclusion is made.

12. Figure 1: Each line on this grey scale figure is very difficult to distinguish. Perhaps the authors may want to use a color figure for better illustration. Also, please add tick marks of each axis. If possible, data obtained at Year 2 & Year 4 could also be added.

13. As a psychophysics test, the results of microperimetry may be significantly affected by the examinees’ cognitive status, which is highly associated with older age and systemic health. It is not surprising that some older subjects get weaker and less cognitive during the 5-year follow-up period even though they are perfectly healthy at the time of enrollment. That’s why an age-matched control group is necessary. The lack of a control group should be discussed as a limitation of this study.

14. Minor comments:

• Careful language polish may be needed. For example, in multiple places (e.g., Line 190, Line 233, Line 234 …) “for” 5 years should be replaced by “during the”; Line 230 “up to the 5-year follow-up” should better be replaced by “during the 5 years”; Line 236-237, “after 5 years” could be “The speckled FAF pattern had the lowest retinal sensitivity compared to the baseline at Year 5/at the 5th year”. Line 245, “at 5 years” should be “at the 5th year”. Line 246, “after 3 years” should be “after the 3rd year”. The expression of “The speckled FAF pattern at 5 years was the least sensitive” should be “The speckled FAF pattern at the 5th years had the lowest mean retinal sensitivity”. Line 252, “more” should be “better”. There are more similar errors, the authors may need to carefully check it thoroughly.

• Caption of Figure 2: there is no data are expressed as mean ± standard error, in the figure.

• Line 198-199: need reformatting.

• Line 205-205: Polypoidal choroidal vasculopathy, wet AMD, and retinal angiomatous proliferation were detected in 4, 9, and 1 eyes, respectively (Table 1). But Table 1 didn’t contain such information.

• Line 209: lost should be loss.

• Line 209-210, “The number of patients lost to follow-up was 2, 5, 6, 3, and 3 at the 1-, 2-, 3-, 4-, and 5-year follow-ups”, please move this sentence to Line 191, after “Of the remaining 191 eyes”, in order to minimize any confusion.

• Line 310, “after” 5 years may be inaccurate.

Reviewer #3: General comments

The authors report on a comparison between microperimetry and FAF patterns in a cohort of patients followed longitudinally for five years. Several are lost to follow-up or convert to CNV in the fellow eye, leaving 27 total eyes that were followed fully for five years. The authors show that test patches ‘close’ to abnormal areas seen on FAF decrease over three years. The authors show that one FAF pattern ‘speckled’ significantly decreased between both 3 and 5 years from baseline. Unfortunately, they only had one participant that exhibited this pattern, making it difficult to make generalizable conclusions based on a single observation. Another major drawback is that the authors only looked at FAF and did not assess how other pathology like large drusen at the test patch locations could have altered sensitivity in those areas, especially as drusen are dynamic and could change over the five years. The manuscript could be improved by better weighting of this finding with respect to the rest of the results as it seems like there is too much emphasis placed on this point. It is also difficult to understand how the FAF image classification changed over time because the authors state that for 14/27 followed for five years that they changed over time but it doesn’t mention how they changed from year to year – this is particularly confusing with respect to figure 1 as this figure implies that the pattern stayed the same over time. This point needs to be clarified. There are also several specific items that could be altered to improve the overall presentation of the work, described below.

Specific comments

Abstract, lines 60-62: The wording of the descriptions of the classification of test points is awkward, please consider rephrasing.

Abstract, lines 69-71: This is confusing as written. Please rephrase from “…lost statistical significance…” to make this clearer.

Abstract, lines 72-73: The emphasis here on the speckled pattern relies on an n=1, perhaps the authors can add some additional points to this conclusion statement to less heavily weight this finding based on a single eye.

Introduction, line 77: The phrasing “…in the elderly in developed countries.” Suggests that AMD only exists in developed countries, please rephrase.

Introduction, line 78: Some classification systems designate AMD as being AMD only if it is in persons >55 years of age, the authors should state the AMD classification system they are using here (e.g. Beckman, AREDS, etc.).

Introduction, line 84: Again, state the scale you are using to define intermediate AMD, etc.

Introduction, line 84-87: This entire sentence needs to be re-written as it is difficult to follow. Certain classification systems outline clearly what constitutes early, intermediate and advanced AMD, the authors should pick one to use and then clearly state the differences between the different stages here.

Introduction, line 91: As written, this is awkwardly phrased, particularly “…overweight, and genetic factors are considerable.”

Introduction, line 95: It is unclear what the authors are intending to state here as microperimetry give a functional change, not a ‘fundus change’. Please rephrase or reword this sentence.

Introduction, lines 100-101: This sentence does not seem to add anything and can be omitted.

Methods, lines 169-170: Who did the image grading? Was it always the same person? What criterion did they use to demarcate the areas manually? Were comparisons made between graders. It would be important for you to include enough detail here for someone else to replicate this experiment.

Results, general: The order the results are presented in does not make sense. The discussion of the transition between

patterns (lines 265-269) needs to be described in better detail and should come before the presentation of the (lines 232-239) of three- and five-year findings.

Results, line 192: Replace “…autofluorescent examination…” with “…FAF imaging…”.

Results, line 198: This sentence appears on its own and should be a part of the preceding paragraph. It should be restated here that only 27 eyes were followed for 5 years – that is how you get 44% from just 12 eyes.

Results, line 199-201: This sentence also could be merged into the preceding paragraph – it doesn’t need to be its own paragraph.

Results, lines 233-239: Are these results for participants patterns only at the last timepoint? How did they change from the first timepoint, was there a difference between those whose patterns changed and those that stayed the same?

Results, 266-269, 279-281: This is confusing and needs to be completely rewritten. The authors may need to make a table or something describing the different transitions between different patterns. It cannot be understood fully from this prose how the patterns of 66 eyes changed over 5 years.

Discussion, line 292: The word “also” can be omitted from this sentence.

Discussion, lines 299-300: What is meant by the “low-fluorescent portion”? I think we need to see these images.

Discussion, lines 300-302: Do the authors mean progress to GA? They all have dry AMD so this does not make sense.

Discussion, line 304: It seems like the authors have the data to answer this question. Are the test patches that were ‘close’ now ‘within’ abnormal areas at year 3 and is this why they have the same sensitivity as ‘within’ patches from 3-5 years?

Discussion, line 311: Do the authors mean that it is a risk factor for AMD progression?

Discussion, line 313: What is meant by ‘…consistent with pigmentation…”?

Discussion, line 316: We need more information on what the patterns transitioned to so that this point can be better understood. Did the lace pattern transition to a less altered pattern?

Discussion, line 317: Do the authors mean “areas” here rather than “lesions”?

Discussion, line 318 & 319: Do the authors mean “test patches” here rather than “lesions”?

Discussion, line 325: The authors should state ‘…”Close” test points…” rather that ‘…”Close” patterns…’

Conclusions, general: This section needs to be completely rewritten, there is too much emphasis on the speckled pattern finding and there are some points that don’t make sense (see specific comment below).

Conclusions, line 330: These patients all have dry AMD – see comment on this above.

Figure 1: Are these results for subjects whose patterns were always consistently the same at each timepoint? It is unclear what the number of subjects is in each pattern at each timepoint – maybe that could be included? Why do we have only 3 and 5 year timepoints shown here but we see sensitivity at a more granular level compared in figure 2?

Figure 2: The figure caption needs to be improved so it is clearer what is being tested for significance here at each timepoint (or if it is being tested to the first timepoint?). The (*) is compared to baseline timepoint but the (#) is compared to close at the same timepoint or at baseline? I think that a bar chart with lines showing comparisons may be needed to show exactly what was tested here. I think it would be useful for the authors to state whether the ‘close’ points at baseline would be graded as ‘within’ at the year 3 timepoint – it seems to be what is implied here. Are these test patches just now within abnormal areas or are they still ‘close’?

6. PLOS authors have the option to publish the peer review history of their article (what does this mean?). If published, this will include your full peer review and any attached files.

Reviewer #1: No

Reviewer #2: No

Reviewer #3: No

---

## [Author Response · Author response to Decision Letter 0]

24 Jan 2020

We appreciate the comments from all reviewers. We carefully responded to all of comments raised by three reviewers. Now we believe that all issues are addressed in the revised version of the manuscript. The part of sentences changed is written in red. We reanalyzed our results according to the reviewers’ questions. Therefore, we added new author (Takeshi Nishiyama) who analyzed the results. I would be pleased to respond to any questions or comments you may have. We re-wrote the title and introduction for better understanding this manuscript. NOTE: The order of results, references and figures have been changed.

---

## [Editor Report · Decision Letter 1]

12 Feb 2020

Five-Year Follow-Up of Fundus Autofluorescence and Retinal Sensitivity in the Fellow Eye in Exudative Age-Related Macular Degeneration in Japan

PONE-D-19-28095R1

Dear Dr. Yasukawa,

We are pleased to inform you that your manuscript has been judged scientifically suitable for publication and will be formally accepted for publication once it complies with all outstanding technical requirements.

With kind regards,

Yuhua Zhang

Academic Editor

PLOS ONE
---

## [Editor Report · Acceptance letter]

24 Feb 2020

PONE-D-19-28095R1 

Five-Year Follow-Up of Fundus Autofluorescence and Retinal Sensitivity in the Fellow Eye in Exudative Age-Related Macular Degeneration in Japan 

Dear Dr. Yasukawa:

I am pleased to inform you that your manuscript has been deemed suitable for publication in PLOS ONE. Congratulations! Your manuscript is now with our production department. 

With kind regards,

on behalf of

Dr. Yuhua Zhang 

Academic Editor

PLOS ONE